# Daily association between perceived control and resolution of daily stressors strengthens across a decade of adulthood

Dakota D. Witzel [1,2,10], Eric S. Cerino [3,4,10] ✉, Robert S. Stawski[5], Gillian Porter[4], Amanda D. Black[3], Raechel A. Livingston[3], Jonathan Rush[6], Jacqueline Mogle[7], Susan T. Charles[8], Jennifer R. Piazza[9] & David M. Almeida[2]

Greater perceived control is often associated with better responses to life's stressors. One reason for this link may be that greater perceived control is related to the ability to resolve these stressful experiences. Using longitudinal data from the National Study of Daily Experiences (N = 1766, Mage = 56.25, SD = 12.20, 57% women, 43% men), we examined associations between perceived control over daily stressors and the likelihood of stressor resolution, and how associations varied over a decade. In two waves conducted in ~2005 and ~2015, participants reported perceived control and resolution of their daily stressors across eight consecutive days. Generalized multilevel models adjusted for trends across days and waves, as well as number of stressors, gender, education, and race. People experiencing greater stressor control across the study days were more likely to report stressor resolution (OR = 1.92, 95%CI: 1.74–2.13, p < 0.001). Further, individuals were more likely to report stressor resolution on days when they reported greater control over their stressors than usual (OR = 1.66, 95%CI: 1.57–1.77, p < 0.001). This within-person association increased in magnitude across waves (OR = 1.21, 95%CI: 1.06–1.39, p < 0.01), resulting in a stronger association between stressor control and resolution when individuals were 10 years older (OR = 1.89, 95%CI: 1.69–2.12, p < 0.001). Results indicate perceived control is a psychosocial correlate of stressor resolution and an important appraisal resource for daily stress processes across the adult lifespan.

## Daily association between perceived control and resolution of daily stressors strengthens across a decade of adulthood

Perceived control, the extent to which one believes that their actions can evoke desirable outcomes, is an important psychosocial resource for health and well-being outcomes (e.g.,[1,2]). Perceived control may benefit well-being in several ways, including how we navigate our daily stressors. Emerging research suggests that perceived control over daily stressors is a domain-specific type of control that may serve as a vital resource for socioemotional experiences across the adult lifespan[3,4]. One reason why greater perceived

control may promote well-being is through its link to the resolution of daily stressors. Resolution of stressors (e.g., an argument settled) is emerging as a characteristic of the daily stress process that is crucial for emotional downregulation (e.g., reduced affective reactivity and residue[5]). In the current study, we examine whether perceived control over daily stressors may be associated with increased likelihood of daily stressor resolution.

The daily stress process model[6] defines the experience of daily stressors (i.e., minor but frequent hassles that occur from normal day-to-day living) as a dynamic process that involves stressor characteristics (e.g., frequency of stressor exposures, diversity of different types of stressors), stressor

[1]School of Education, Counseling, and Human Development, South Dakota State University, Brookings, SD, USA. [2]Center for Healthy Aging, Pennsylvania State University, University Park, PA, USA. [3]Department of Psychological Sciences, Northern Arizona University, Flagstaff, AZ, USA. [4]Interdisciplinary Health Program, Northern Arizona University, Flagstaff, AZ, USA. [5]Department of Human Development and Family Studies, Utah State University, Logan, UT, USA. [6]Department of Psychology, University of Victoria, Victoria, BC, Canada. [7]RTI Health Solutions, Durham, NC, USA. [8]Department of Psychological Science, University of California Irvine, Irvine, CA, USA. [9]Department of Public Health, California State University, Fullerton, CA, USA. [10]These authors contributed equally: Dakota D. Witzel, Eric S. Cerino. ✉e-mail: Eric.Cerino@nau.edu

appraisals (e.g., control, resolution), and stressor responses (i.e., emotional, behavioral, or biological reactions to stressors on the same day known as reactivity or prolonged responses extending to the following day known as residue). Together, these components are the micro-level processes through which proximal daily experiences contribute to distal long-term health and well-being across adulthood.

Research on daily stress processes[7–10] predominately examines exposures and affective responses to daily stressors (e.g., negative affective reactivity); however, recent calls to expand and integrate components of the daily stress process model (e.g., [6]) look to build on emerging research on the roles of comparatively less studied stressor appraisals, including stressor control and resolution (see Fig. 1 for conceptual model highlighting appraisals in the daily stress process). In line with Lazarus and Folkman's[11] seminal work on stress and coping theory, appraisals are crucial but underutilized aspects of stress processes – particularly at the daily level. Stressor appraisals theoretically[6] and empirically[5] modify reactivity to daily stressors, but few studies have explored (1) whether aspects of daily stressor appraisals relate to each other and (2) the extent to which daily stressor appraisals change across adulthood. For example, feeling control over one's stressors may increase the ability to manage and resolve a daily stressor, thus optimizing reactions to daily stressors and promoting daily health and well-being. As such, the current study provides an extension of the daily stress process model with formal evaluation of associations between control and resolution of daily stressors.

Accumulating evidence indicates individual differences[12] and daily dynamics (e.g.,[13]) in perceived control can play important roles in daily stress processes. Results from daily diary studies across 30 days[14] and 9-weeks[13] showed that affective reactivity to daily stressors was lower on days people had higher levels of perceived control. Notably, associations between daily control and reactivity to daily stressors may be through other appraisals of stress, such as resolution. For example, perceiving control over daily stressors could help people resolve their stressors by motivating them to address the issue (e.g., feeling control over a billing error may motivate a person to call the company in question to fix it). As such, the present study extends the current line of work by evaluating the association between daily stressor control and the resolution of daily stressors.

Daily stressor control, the sense that people feel individual agency over daily life challenges, is distinct from global levels of control in broad domains of life, such as interpersonal tensions and overloads at work and home[4,15]. People remain stable in their perceived control over daily interpersonal stressors (i.e., arguments and avoided arguments) but decline in their control over non-interpersonal stressors (i.e., work and home demands) across 10 years of follow-up[3]. This preservation of control over interpersonal stressors is consistent with socioemotional selectivity theory, given older adults often work toward optimizing their social relationships[16].

Age patterns in other domains of control in everyday life are more nuanced [c.f., [4]]. General perceived control tends to increase throughout younger adulthood and remain relatively stable in midlife before declining in older adulthood[15,17,18]. However, older age has also been related to different levels of control dependent on the domain (i.e., greater control for work and finances; less control for relationships with children[19]). Determining whether the associations between daily stressor control and resolution differ across ages and as adults get older, therefore, is an important consideration for the current research. Indeed, daily stress research shows people in midlife often report more daily stressors and perceive their stressors as more severe than older adults[20]. Further, coordinated analysis of intensive repeated measurement studies shows age-related reductions in stressor reactivity[9]. Links between control and resolution of daily stressors, then, may change in magnitude as people grow older and shift in their priorities, activities, and resources[17].

Coping literature examines individual differences in the likelihood of resolution for significant life events, with past work identifying directed coping, social resources, and better health as correlates of resolution over up to 10 years of follow-up (e.g, [21,22]). The operationalization of stressor resolution varies widely across studies, from reporting the exact date (year/

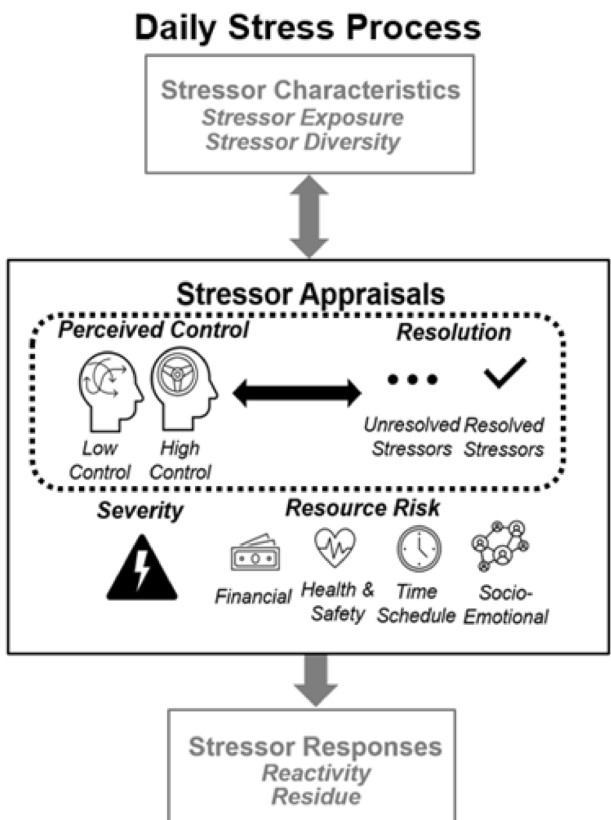

**Fig. 1 | Schematic diagram on associations between control and resolution of daily stressors embedded in the daily stress process model.** The dotted box around the control and resolution components (middle) communicates our emphasis on these two stressor appraisals in the current study within the larger daily stress process model.

month) of the exposure and concrete end[22] to a multi-indicator item determining conflict resolution[23] to a dichotomous Yes/No response for resolution of specific daily stressors at the end of each day of daily diary[5]. Although previous literature has explored how life stressors, for example, tend to resolve across a median of ~7 months, transition stressors were resolved in the shortest amount of time, followed by interpersonal, illness, and role strain stressors[22]. Given this variability in stressor resolution by type, the present study extends this past work to evaluate resolution across *daily* domains of interpersonal, work, home, and network stressors.

Resolution of daily stressors is operationalized as the subjective appraisal that a daily stressor is no longer ongoing[5]. This stressor appraisal may be particularly beneficial for the downregulation of emotions[22], with recent work demonstrating decreases in affective responses when daily stressors are resolved[5]. Research on conflict resolution has primarily focused on indicators and correlates of resolution in the context of marital conflict (e.g., [24]), with fewer studies exploring how aspects of *daily stressors* may promote resolution. For example, in marital conflict literature, research has noted that self-efficacy may be crucial to resolving marital conflict[25]. It remains to be seen whether aspects of control in a broader context (e.g., daily stressors across domains) is related to resolution. Notably, Witzel and Stawski[5], leveraging the adult lifespan sample of the National Study of Daily Experiences (NSDE), demonstrated that affective reactivity and residue associated with interpersonal stressors was attenuated or even extinguished when stressors were resolved[5], suggesting the importance of resolution for daily well-being. Moreover, they examined age differences in stressor resolution as well and found that older adults reported more interpersonal stressor resolution than comparatively younger adults. The current study aims to expand on the current literature to test how stressor control may be associated with the likelihood of stressor resolution across several different

daily stressor domains (e.g., work, home, interpersonal) and age differences therein.

The current study uses an intensive longitudinal design to examine patterns in associations among two often overlooked daily stressor appraisals (i.e., control, resolution). Figure 1 provides a schematic diagram embedding associations between control and resolution within the daily stress process model[6]. Understanding associations between daily stressor appraisals may provide critical information for the development and use of daily stressor interventions across adulthood. We apply an intraindividual variability approach[26] to the study of these dynamic constructs to examine daily associations between stressor control and stressor resolution within persons over time and at the level of individual differences. Although our study design is correlational, we conceptualize stressor resolution as the target outcome and stressor control as the focal predictor to identify a potentially modifiable psychosocial correlate (i.e., control) of a meaningful outcome at the end of a day (whether a stressor has been resolved or not). Indeed, given resolution's relevance for emotional downregulation in past research (e.g.,[5,22]), it is important to identify daily factors that may be correlated with increased likelihood of resolving one's stress. Further, past research has similarly examined resolution as an outcome regressed on psychosocial predictors (e.g.,[21]). Thus, we address the following three research questions:

1. How are control and resolution of daily stressors associated both within-persons over time and between-persons? Within persons, we hypothesize that individuals will be more likely to resolve their stressors on days when their stressor control is higher than usual. Between persons, we hypothesize that individuals with more stressor control on average will be more likely to resolve their stressors.
2. How are associations between control and resolution of daily stressors moderated by baseline age? We hypothesize that the positive association between control and resolution will be stronger among comparatively older adults.
3. How are associations between control and resolution of daily stressors moderated by assessment wave? We hypothesize that control-resolution associations will strengthen across 10 years of follow-up.

## Methods

### Participants and Procedure
We used data from the National Study of Daily Experiences (NSDE), a daily diary subproject of a random subset of participants from the larger Midlife in the United States Survey project. Participants completed daily telephone interviews for eight consecutive days that assessed exposure to daily stressors (for detailed description of data collection, see ref. 27). Data collection consisted of three waves of daily assessments repeated approximately every 10 years (NSDE 1: ~1995; NSDE 2: ~2005; NSDE 3: ~2015). Reported resolution status and perceived control over daily stressors were collected at the second and third waves, resulting in longitudinal daily diary data on control and resolution of daily stressors across 10 years. The analytic sample included 1778 adults (7788 assessments) who participated in Wave 2 and/or Wave 3 of the NSDE (764 participants from Wave 2 also contributed Wave 3 data) and reported at least one daily stressor and thus have data regarding daily stressor resolution and daily stressor control.

### Transparency and openness
Data are publicly available at: https://www.icpsr.umich.edu/web/ICPSR/series/203. All analyses were completed using SAS 9.4[28]. Syntax for study analyses is included in supplementary material. This study was not preregistered. This study was approved by the Institutional Review Board of Pennsylvania State University, and all respondents consented to their participation. Respondents received $25 as compensation for their participation.

### Measures

**Daily stressor control**. Perceived control over daily stressors was assessed as part of the Daily Inventory of Stressful Events (DISE, e.g.,

ref. 29). Participants responded to a series of stem questions asking whether certain types of daily stressors had occurred in the past 24 hours (arguments, avoided arguments, work stressors, home stressors, network stressors). For each stressor reported (occurring on 40% of all available days in Wave 2, 39% of all available days in Wave 3), participants answered the question, "How much control did you have over the situation?" on a 4-point Likert-type scale (0 = *none at all*, 1 = *a little*, 2 = *some*, 3 = *a lot*). Higher values indicate greater perceived control over daily stress. Daily stressor control was obtained by taking the average amount of control over the reported stressor(s) for each of the eight days.

**Daily stressor resolution**. For each stressor reported, participants were asked a direct follow-up question where they endorsed whether the stressor was resolved by answering the question, "Is the situation resolved?" with 0 (*no*) or 1 (*yes*). As such, resolution referred to a subjective appraisal that each participant interprets for themselves when characterizing the stressor they experienced (e.g., the participant appraises their stressor as resolved and answers the interview question with "Yes"). In the current study, daily stressor resolution was operationalized as a dichotomous outcome variable indicating whether any stressor was reported as resolved on each day (0 = *unresolved stressors*, 1 = *at least one resolved stressor*).

**Age**. Chronological age was utilized as a moderator for associations. Baseline age was calculated by subtracting the year participants completed wave 2 from their year of birth. Age at baseline was centered at the sample mean at Wave 2 (i.e., 58 years) in all models.

**Covariates**. Women tend to report lower levels of perceived control than men on average, although these gender differences may be attenuated among adults with a college education[19]. Higher education has been linked to higher levels of perceived control[30]. Further, past work on racial differences in control beliefs suggests that Black/African American individuals may have lower levels of perceived control than white individuals[31], due in part to structural discrimination[32]. Therefore, education, gender, and race were included as covariates in primary analyses with information provided by participants. Education was coded as 0 (*high school or less*) or 1 (*some college or more*). Gender was coded as 0 (*men*) or 1 (*women*). Race was coded as 0 (*white*) or 1 (*nonwhite*). Low cell sizes of individual racial identities of Black/African American (Wave 2: 12%; Wave 3: 4%), Native American or Alaska Native (Wave 2: 1%; Wave 3: 1%), Asian or Pacific Islander (Wave 2: 1%; Wave 3: 1%), and other (Wave 2: 2%; Wave 3: 5%) led to an analytic decision to collapse categories into a not white group in a dichotomous variable. We recognize, however, that the lives of minoritized or historically marginalized adults cannot be equated and do not reflect the same lived experiences across or within racial identities. To adjust for differential exposure to stressors, we covaried for the sum of reported stressors each day.

### Analytic strategy
We used generalized linear mixed models (binary outcome distribution and logit link function in PROC GLIMMIX)[28] to address the research questions. Maximum likelihood estimation was used due to missing data and attrition across days and waves of assessment. Statistical tests were two-sided. Multi-level models (MLMs) had three levels of analysis: days of stressor resolution (level 1) nested within measurement waves (level 2) nested within people (level 3). Intraclass correlation coefficients (ICCs) from unconditional mixed linear models were used to determine within- and between-person variation in primary study variables. Day-level stressor control was computed by subtracting an individual's average level of stressor control from their daily scores[33]. Associations between control and resolution were assessed with three-level generalized MLMs

described below.

$$\text{Level} - 1(\text{day}) : \text{Resolution}_{ijk} = \pi_{0ij} + \beta_1 ij(\text{Day}_{ijk}) + \beta_2 ij(\text{Number of Stressors}_{ijk}) + \beta_3 ij(\text{Control}_{ijk})$$

$$\text{Level} - 2(\text{wave}) : \pi_{0ij} = \delta_{00i} + \delta_{01i}(\text{Wave}_{ij}) + r_{0ij}$$
$$\beta_{3ij} = \delta_{30i} + \delta_{31i}(\text{Wave}_{ij})$$

$$\text{Level} - 3(\text{person}) : \delta_{00i} = \gamma_{000} + \gamma_{001}(\text{Gender}_i) + \gamma_{002}(\text{College}_i) + \gamma_{003}(\text{Race}_i) + \gamma_{004}(\text{BaselineAge}_i) + \gamma_{005}(\text{Control}_i) + \gamma_{006}(\text{Control}_i * \text{Baselineage}_i) + u_{00i}$$
$$\delta_{01i} = \gamma_{010} + \gamma_{011}(\text{Control}_i) + u_{01i}$$
$$\delta_{30i} = \gamma_{300} + \gamma_{301}(\text{BaselineAge}_i)$$

The odds of resolving at least one stressor for person $i$ at wave $j$ on day $k$ is a function of level-1 intercept ($\pi_{0ij}$), linear trend across days ($\beta_1 ij$), number of stressors reported that day ($\beta_2 ij$), within-person stressor control ($\beta_3 ij$), level-2 linear trend across waves of assessment ($\delta_{01i}$), level-3 between-person differences in gender ($\gamma_{001}$), education ($\gamma_{002}$), race ($\gamma_{003}$), age at baseline ($\gamma_{004}$), and person-means of stressor control ($\gamma_{005}$), as well as random effects for the intercept at level-2 ($r_{0ij}$), and the intercept ($u_{00i}$) and wave of assessment ($u_{01i}$) at level-3 to allow for variation across persons and waves. To answer research question 1 (How are control and resolution of daily stressors associated both within-persons over time and between-person?), we regressed daily stressor resolution on within-person stressor control ($\beta_3 ij$) and between-person stressor control ($\gamma_{011}$).

To answer research question 2 (How are associations between control and resolution of daily stressors moderated by baseline age?), we added two interaction terms. Specifically, we added the within-person stressor control*baseline age two-way interaction ($\gamma_{301}$) and the between-person stressor control*baseline age two-way interaction ($\gamma_{006}$) as predictors of stressor resolution in Model 2.

To answer research question 3 (How are associations between control and resolution of daily stressors moderated by assessment wave?), we included two additional interaction terms. Specifically, we added the within-person stressor control*wave of assessment two-way interaction ($\delta_{31i}$) and the between-person stressor control*wave of assessment two-way interaction ($\gamma_{011}$) as predictors of stressor resolution in Model 3.

We report odds ratios (OR[34]) with 95% confidence intervals (CI) to indicate the difference in odds of resolving at least one stressor for a 1-unit increase in stressor control compared to average control. To further aid in the interpretability of effects, we also calculated the predicted probability ($\frac{Odds}{1+Odds}$) of resolving at least one stressor given a 1-unit increase in stressor control compared to average control.

## Results

Tables 1 and 2 provide descriptive statistics and bivariate correlations for primary study variables across waves of assessment, respectively. On average, people perceived *a little* to *some* control over their stressors at Wave 2 (Mean=1.45) and Wave 3 (Mean=1.44), with 69% and 68% of stressors reported as resolved at Wave 2 and Wave 3, respectively. Unconditional models showed significant between-person and within-person variation in measures of daily stressor control and daily stressor resolution (Fig. 2).

Bivariate correlations were evaluated using the person mean for average stressor control across stressor days, person mean for total number of stressors reported each day, and the percentage of days when at least one stressor was resolved across stressor days. At Wave 2, significant bivariate correlations with covariates showed that higher stressor control was reported more by men (compared to women; $r = -0.11$, $p < 0.001$), non-white respondents (compared to white respondents; $r = 0.09$, $p < 0.001$), and people with fewer numbers of stressors on stress days ($r = -0.06$, $p < 0.01$). Further, stressor resolution at Wave 2 was significantly related to education and number of stressors such that people with less than some college

## Table 1 | Descriptive statistics by wave of assessment

| Variable | Wave 2 | | Wave 3 | |
| --- | --- | --- | --- | --- |
| | M(SD)/% | Range | M(SD)/% | Range |
| 1. Age | 58.61 (12.12) | 35–86 | 67.67 (10.34) | 47–95 |
| 2. Women | 57% | 0–1 | 57% | 0–1 |
| 3. Race (non-white %) | 16% | 0–1 | 11% | 0–1 |
| 4. College (%) | 69% | 0–1 | 77% | 0–1 |
| 5. Number of stressors | 1.26 (0.38) | 1–4 | 1.22 (0.33) | 1–4 |
| 6. Stressor control | 1.45 (0.86) | 0–3 | 1.44 (0.79) | 0–3 |
| 7. Stressor resolution (%) | 69% | 0–1 | 68% | 0–1 |

*N = 1778 at wave 2, 1236 at wave 3. Women (0 = men, 1 = women). College (0 = high school or less, 1 = some college or more). Race (0 = white, 1 = non-white). Number of stressors = person mean for the total number of stressors reported each day. Stressor Control = person mean for average stressor control across stressor days. Stressor Resolution = Percentage of days when at least one stressor was resolved across stressor days.*

($r = -0.09$, $p < 0.001$) and fewer numbers of stressors on stress days ($r = -0.07$, $p < 0.01$) reported higher percentage of days when at least one stressor was resolved. At Wave 3, significant bivariate correlations showed stressor control was reported more by men (compared to women; $r = -0.10$, $p < 0.001$). Further, stressor resolution at Wave 3 was significantly related to education and number of stressors such that people with less than some college ($r = -0.11$, $p < 0.001$) and fewer numbers of stressors on stress days ($r = -0.06$, $p < 0.05$) reported higher percentage of days when at least one stressor was resolved.

### Control Associated with Odds of Resolving Stress
**Main effects.** Within persons, reporting greater stressor control than usual was associated with increased stressor resolution (OR = 1.66, 95% CI: 1.57–1.77, $p < 0.001$; Table 3, Model 1; Fig. 3). In other words, individuals were more likely to report resolving stressors on days when they perceived more control over their stressors than usual. In terms of predicted probability per 1-unit difference in within-person stressor control (e.g., *none at all* to *a little* control), the probability of resolving stressors on days when stressor control was higher than usual was 0.62. Between persons, individuals with more stressor control on average were more likely to resolve their stressors (OR = 1.92, 95% CI: 1.74–2.13, $p < 0.001$; Table 3, Model 1; Fig. 3). In terms of predicted probability per 1-unit difference in between-person stressor control, the probability of resolving stressors for people with more stressor control on average was 0.66.

**Moderation by age differences.** Cross-sectionally, baseline age did not significantly moderate associations between stressor control and stressor resolution ($ps > 0.45$; Table 3, Model 2).

**Longitudinal moderation.** Estimates for longitudinal changes in the association between stressor control and stressor resolution are provided in Table 3 (Model 3) and Fig. 4. The within-person association significantly increased in magnitude across assessment waves (OR = 1.21, 95% CI: 1.06–1.39, $p < 0.01$), resulting in a stronger association between stressor control and resolution 10 years later (OR = 1.89, 95% CI: 1.69–2.12, $p < 0.001$) compared to daily associations at baseline (OR = 1.56, 95% CI: 1.45–1.68, $p < 0.001$). We did not find statistically significant evidence for longitudinal moderation of between-person associations among stressor control and resolution across the 10-year follow-up. In terms of predicted probability per 1-unit difference in within-person stressor control, the probability of resolving stressors on days when stressor control was higher than usual was 0.65 ten years later compared to 0.61 at baseline.

### Table 2 | Bivariate correlations by wave of assessment

| | 1 | 2 | 3 | 4 | 5 | 6 | 7 |
|---|---|---|---|---|---|---|---|
| 1. Age | — | −0.02[−0.08, 0.03] | −0.001[−0.06, 0.06] | −0.09*[−0.15, −0.03] | −0.18***[−0.24, −0.13] | −0.05†[−0.11, 0.01] | −0.01[−0.07, 0.04] |
| 2. Women | −0.02[−0.07, 0.02] | — | .03[−0.03, 0.09] | −0.11**[−0.17, −0.05] | .10***[0.04, 0.15] | −0.10**[−0.16, −0.04] | −0.03[−0.09, 0.03] |
| 3. Race (non-white %) | −0.05[−0.10, −0.01] | 0.05[0.01, 0.10] | — | 0.001[−0.06, 0.06] | −0.06*[−0.12, −0.001] | −0.02[−0.09, 0.04] | −0.01[−0.07, 0.05] |
| 4. College (%) | −0.11***[−0.15, −0.07] | −0.07†[−0.11, −0.03] | −0.10**[−0.14, −0.05] | — | .17***[0.11, 0.23] | −0.02[−0.08, 0.04] | −0.11***[−0.18, −0.05] |
| 5. Number of stressors | −0.24***[−0.28, −0.19] | .08**[0.04, 0.12] | −0.04†[−0.08, 0.01] | .18***[0.13, 0.22] | — | −0.01[−0.07, 0.05] | −0.06*[−0.12, −0.01] |
| 6. Stressor control | −0.02[−0.06, 0.03] | −0.11***[−0.16, −0.07] | .09***[0.05, 0.14] | −0.04†[−0.09, 0.01] | −0.06*[−0.11, −0.02] | — | 0.28***[0.23, 0.33] |
| 7. Stressor resolution (%) | −0.02[−0.07, 0.02] | −0.03[−0.07, 0.02] | 0.03[−0.02, 0.08] | −0.09***[−0.14, −0.04] | −0.07**[−0.11, −0.02] | 0.30***[0.26, 0.34] | — |

*N* = 1778 at wave 1, 1236 at wave 2, 1236 at wave 3. Women (0 = *men*, 1 = *women*). Race (0 = *white*, 1 = *non-white*). College (0 = *high school or less*, 1 = *some college or more*). Stressor Resolution = Percentage of days when at least one stressor was resolved across stressor days. Stressor control = person's mean for average stressor control across stressor days. Number of stressors = person mean for the total number of stressors reported each day. Stressor Control = person's mean for average stressor control across stressor days. Bivariate correlations at wave 2 are reported below the diagonal with 95% confidence intervals in brackets. Bivariate correlations at wave 3 are reported above the diagonal with 95% confidence intervals in brackets. †*p* < 0.10. *p < 0.05. **p < 0.01. ***p < 0.001.

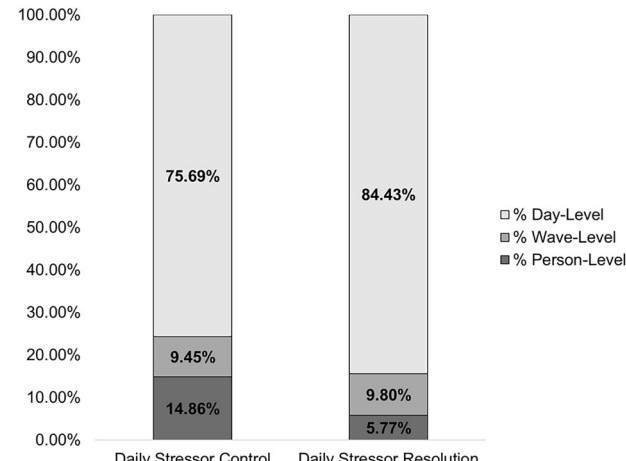

**Fig. 2 | Variance decompositions for primary study variables.** Values depicted reflect proportion of variation across persons, waves, and days. *N* = 1,778 participants, 7788 observations.

## Sensitivity analysis

We conducted four sets of sensitivity analysis to evaluate the robustness of the present findings. Full model results for each set of sensitivity analysis are provided in Supplementarymaterial.

**Days when only one stressor was reported.** We determined whether findings operated similarly on days when only one stressor was reported (Supplementary Table 1). Results of this sensitivity analysis revealed that primary findings held when reducing the analytic sample to days when only one stressor was reported. Resolution was more likely to be reported on days when people perceived more control over their stress than usual and for people that reported more stressor control on average. Consistent with primary analysis, we did not find statistically significant evidence for baseline age moderation, and the within-person association significantly increased in magnitude across assessment waves.

**Adjusting for severity of daily stressors.** Due to the potential for the severity of stressors to be related to the controllability of the stressor and its impact on the stress process (e.g.,[35]), we reran each model covarying for stressor severity reported on each stressor day (see Supplementary Table 2). Results revealed that primary findings held when additionally adjusting for the severity of daily stressors.

**Adjusting for education level.** We determined whether findings operated similarly when covarying for education level (range = 1–12) instead of the dichotomous education covariate used in primary analysis (see Supplementary Table 3). Results revealed that primary findings held when covarying for education level instead of the dichotomous covariate.

**Different types of stressors.** Fourth, we conducted sensitivity analysis to determine whether daily links and individual differences operated similarly across different types of stressors. Supplementary Table 4 includes results from models examining associations in arguments, avoided arguments, work stressors, home stressors, and network stressors. Results of these sensitivity analyses revealed that within- and between-person associations tended not to vary as a function of stressor type. For additional information on resolution across stressor types, we included the percentage of days when at least one stressor was resolved across stressor days for each type of stressor in Supplementary Table 5.

**Table 3 | Logistic generalized MLMs for control associated with resolution of daily stressors**

| Parameter | Model 1:<br>Main Effects<br>OR [95% CI] | Model 2:<br>Moderation by Age Differences<br>OR [95% CI] | Model 3:<br>Longitudinal Moderation<br>OR [95% CI] |
|---|---|---|---|
| Fixed Effects | | | |
| Day | 1.01 [0.99, 1.04] | 1.01 [0.99, 1.04] | 1.01 [0.99, 1.04] |
| Number of Stressors | 2.16 [1.92, 2.43] | 2.16 [1.92, 2.43] | 2.15 [1.91, 2.42] |
| Wave | 1.10 [0.95, 1.27] | 1.10 [0.95, 1.27] | 0.92 [0.67, 1.27] |
| Women | 0.97 [0.84, 1.13] | 0.97 [0.84, 1.13] | 0.97 [0.84, 1.12] |
| Race | 0.96 [0.78, 1.19] | 0.96 [0.78, 1.18] | 0.97 [0.84, 1.12] |
| College | 0.68 [0.57, 0.80]*** | 0.68 [0.57, 0.80]*** | 0.68 [0.57, 0.80]*** |
| Age at Baseline | 0.999 [0.99, 1.01] | 0.995 [0.98, 1.01] | 0.999 [0.99, 1.01] |
| WP Stressor Control | 1.66 [1.56, 1.77]*** | 1.67 [1.57, 1.78]*** | 1.56 [1.45, 1.68]*** |
| WP Stressor Control X Age at Baseline | - | 1.002 [0.99, 1.01] | - |
| WP Stressor Control X Wave | - | - | 1.21 [1.06, 1.39]** |
| BP Stressor Control | 1.92 [1.74, 2.13]*** | 1.93 [1.75, 2.13]*** | 1.85 [1.65, 2.06]*** |
| BP Stressor Control X Age at Baseline | - | 1.003 [0.99, 1.01] | - |
| BP Stressor Control X Wave | - | - | 1.18 [0.95, 1.46] |
| Level-3 Random Effects | | | |
| Intercept Estimate (SE) | 0.38 (0.10) | 0.38 (0.10) | 0.39 (0.10) |
| Wave Estimate (SE) | 0.04 (0.18) | 0.03 (0.18) | 0.18 (0.20) |
| Level-2 Random Effect | | | |
| Intercept Estimate (SE) | 0.43 (0.13) | 0.43 (0.13) | 0.36 (0.13) |
| −2LL | 8985.57 | 8984.54 | 8975.94 |

*N* = 1778 participants, 7788 observations. WP=within-person. *BP* between-person. Age at Baseline = centered at sample average age at wave 2 (58 years). WP Stressor Control = within-person deviation scores for stressor control. BP Stressor Control = person-mean values for stressor control. Estimates and standard errors (SE) represent parameter estimates and odds ratio (OR) estimates reflect exponentiated estimates with 95% confidence intervals (CI). †*p* < 0.10. *\*p* < 0.05. *\*\*p* < 0.01. *\*\*\*p* < 0.001.

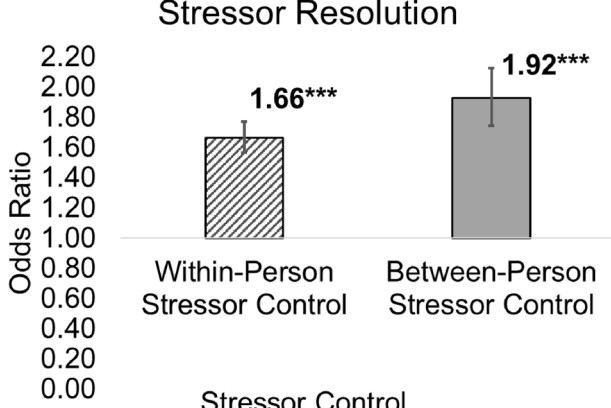

**Fig. 3 | Within- and between-person associations among control and resolution of daily stressors.** ***\*\*\*p* < 0.001. Values listed in the figure reflect odds ratios (bars reflect 95% confidence intervals). *N* = 1778 participants, 7788 observations.

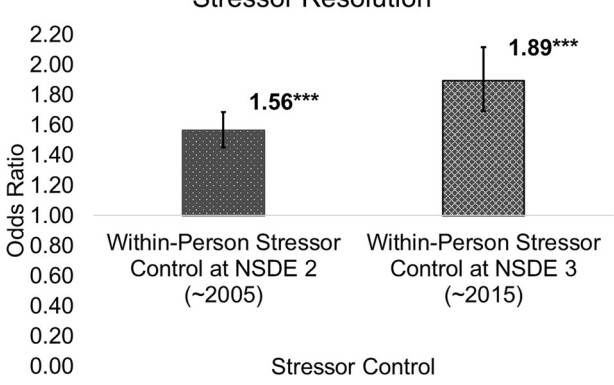

**Fig. 4 | Within-person association increased in magnitude across 10-year follow-up.** ***\*\*\*p* < 0.001. Values listed in the figure reflect odds ratios (bars reflect 95% confidence intervals). *N* = 1778 participants, 7788 observations.

## Discussion

Results from this study indicated that perceived control is a psychosocial correlate of stressor resolution and an important appraisal resource for daily stress processes across the adult lifespan. Longitudinal changes in these within-person associations over a decade of adulthood suggest that leveraging one's daily stressor control to promote stressor resolution may become more salient as people grow older and control resources become more finite (e.g. ref. 17). At baseline, there was a 56% increase in the odds of resolving stress on days when control was one unit higher than average (e.g., feeling a lot of control compared to usual days of feeling some control). Over

time, this effect increased to 89% greater odds of resolving stress on days when control was higher than usual.

### Control and resolution as integrated components of the daily stress process model

The current study informs theory (e.g., [6,11]) and provides an empirical response to recent calls to expand and integrate components of the daily stress process model (e.g., [6]) through formal examination of two under-studied appraisals: control and resolution of daily stressors. Descriptive statistics clarified that control and resolution are correlated, but distinct, appraisals. Although variation in both appraisals predominantly reflected

time-varying sources within persons over time, variation in resolution reflected comparatively more variation across days than control. Resolution may be comparatively more contingent on external factors that vary from day-to-day than control and future work should continue to explore the daily and environmental determinants of resolution.

Stressor resolution was more likely to occur on days when people perceived more control over their stressors, regardless of stressor domain. Previous work shows that higher levels of daily control buffers affective reactivity to daily stressors[13,14]. The current study extends this literature by demonstrating that stressor resolution is an additional component that contributes to this buffering effect. In addition to the within-person associations, individual differences in control over stressors on average were also related to resolution such that people who perceived more control over their stressors on average were also more likely to resolve their stressors. This between-person association is consistent with past work documenting individual differences in the role of perceived control as a buffer against affective reactivity[12] and as a correlate of health outcomes[1,2,36]. Results from the current study clarify that individual differences in stressor control are also useful in understanding who is more likely to resolve their stressors (i.e., people with more perceived control over their stressors).

Although these were expected associations consistent with our hypotheses, it is important to formally demonstrate associations among a modifiable psychosocial resource like stressor control and the likelihood of resolving daily stressors to inform more personalized approaches to healthy aging[37]. Indeed, the strengthening of control-resolution associations over time and consistency in findings across different types of stressors identify the importance of studying control-resolution linkages as people grow older. Future efforts aimed at enhancing daily stress responses by increasing individuals' capacity to resolve stressors may be effective not only for individuals with high levels of stressor control, but also for all adults when their momentary sense of control is elevated.

### Control as a psychosocial correlate of daily stressor resolution

Given resolution's role in initiating the downregulation of emotions[38], it is important to identify modifiable factors in daily life that may contribute to resolving stressors. The current study identifies perceived stressor control as one potentially modifiable psychosocial correlate that may be crucial for promoting stressor resolution within persons over time and at the level of individual differences. The current study used a measurement burst design (i.e., micro-level daily diaries nested within macro-level waves of assessment) coupled with a multilevel modeling approach to examine longitudinal changes and baseline age differences. As a result, we could assess daily associations for whom (i.e., people with more stressor control on average) and the specific daily life contexts (i.e., when people perceive more control over their stress) when stressor control is associated with increased likelihood of stressor resolution.

### Longitudinal changes in the daily control-resolution association.

When examining associations across time, however, there was an aging-related strengthening of control-resolution associations. The change in within-person associations between control and resolution further clarified how short-term processes like control and resolution can change across a distal long-term process. Specifically, the daily link between more stressor control and higher odds of resolution was stronger at 10-year follow-up than it was at baseline. Leveraging one's daily stressor control to promote stressor resolution, then, may become more salient as people grow older and control resources become more finite in later life. People's ability to independently exert control over their environment generally tends to increase throughout younger adulthood, remain relatively stable in midlife, and then decrease in later adulthood[15,17,18]. Control over specific life domains, however, are more nuanced (e.g., for review, see [4]), with older age related to greater control over their work and finances, but less control over domains such as their sex life and relationships with children[19]. Further, certain types of control in daily life such as interpersonal stressor control remain stable across up to a 10-year follow-up[3].

Regarding age differences in resolution, older adults report more interpersonal stressor resolution than comparatively younger adults[5]. Perhaps people became more effective at allocating their perceived control toward stressor resolution as they got older because their priorities and resource allocation shifted toward preserving their well-being in daily life. It may also be the case that people accumulate more expertise and resources in handling daily stressors, thereby experiencing higher levels of control and a stronger relationship with resolution a decade later. These age-related strengths may partially explain why this within-person association strengthened over time.

The strengthening of the control-resolution association over ten years of follow-up may also be due to shifts in social roles and external circumstances that may impact the need for stressor resolution as people grow older. Work and family demands earlier in adulthood may be characterized by more external circumstances (e.g., preparing for an external evaluation at work or helping your parent manage medications before their doctor's appointment). As people grow older and shift into social roles comparatively less-dictated by external circumstances, their environments may allow for more flexibility to resolve the stressors using their own control resources. This potential heightened alignment between fewer external circumstances and greater capacity to allocate control over the stressors that matter to them may also be partially explaining why the within-person control and resolution link grew stronger ten years later.

### Limitations

The strengths of this study must be interpreted alongside its limitations. First, the sample's lack of diversity in racial and ethnic composition, as well as socioeconomic status, is a limitation on generalizability. Importantly, daily stressors are contextual in nature; both racial minorities and people living in poverty (as well as their intersection) face qualitatively different stressors that may result in different resources and ability to perceive control over and/or resolve a stressor. With the population increasingly becoming diverse in socioeconomic, racial, and ethnic composition, it is crucial for future work to include a more diverse sample with additional assessment waves to better elucidate patterns of change across time and evaluate how changes in the association between control and resolution of daily stressors may vary as a function of sociodemographic, personality, and health characteristics.

An important future direction is to examine the potential bidirectionality of associations between control and resolution. The within-person associations in the current study's daily diary design are correlational. Further, it was not possible to evaluate control and resolution over the same stressors across multiple days (respondents reset their reporting of daily stressors each day). Thus, an important limitation of the current study is its inability to evaluate temporal effects of control on resolution of specific stressors from one day to the next. Future research should address this limitation with more momentary assessments within days (e.g., ecological momentary assessment designs) to examine whether increases in control lead to subsequent resolution of stressors and/or if people increase their perception of control in response to resolving a stressor. Disentangling the temporal effects of resources, control, and resolution will become crucial for understanding how and when to intervene in daily stress processes.

### Future directions

The present study focused on an individual's subjective appraisal of whether their stressors were resolved or not. Although we ask about whether the stressor itself has been resolved, this question still raises the possibility that people are thinking about what happened and may have lingering emotions in response. The present study was a first step in assessing resolution. Future research can examine additional features of the resolution process, such as when the stressor was resolved, how it was resolved, and the possible emotional responses associated with the resolution experience. This line of research can inform how resolution status relates to the development of chronic stress. One way to study chronicity of stressor exposures is to assess how often the same domains of stressors are reported (e.g., [39]). Past research shows that the combination of low diversity of stressor exposure and high

levels of stressor exposure in general are associated with elevated stressor reactivity and may reflect the chronicity of daily stressors[39]. Future measurement burst research that incorporates more comprehensive features of the resolution process for these stressor exposures embedded within macro-longitudinal follow-up can evaluate how persistent unresolved stressors over multiple days may accumulate over consecutive days and months to become a chronic strain.

## Conclusion

Perceiving control over daily stressors is a psychosocial correlate of stressor resolution, a component of the daily stress process with relevance for emotional downregulation in daily life. Leveraging one's daily stressor control to promote stressor resolution may become more salient as people grow older and control resources become more finite. As such, the current research uses an intensive longitudinal design to add to the literature by testing the within-person and between-person relationships between two appraisals of daily stressors comparatively understudied in the literature: perceived stressor control and whether the stressor has been resolved. Indeed, the current research is an initial but necessary first step to the examination of fine-grained daily processes that can be utilized to promote health and well-being in daily life and throughout adulthood.

## Data availability

Data are publicly available at: https://www.icpsr.umich.edu/web/ICPSR/series/203[40]. The specific data used to create tables and figures in the present study is available on the open science framework at the following link: https://osf.io/9wuf5/?view_only=668af2adcc3a4b63b5af8ee78832c53d.

## Code availability

All analyses were completed using SAS 9.4. Syntax for study analyses is available on the open science framework at the following link: https://osf.io/9wuf5/?view_only=668af2adcc3a4b63b5af8ee78832c53d.

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

## Acknowledgements

We would like to thank the MIDUS staff, research team, and participants for their time and effort. Research was supported by the National Institute on Aging, under award numbers: P01 AG020166, M3 U19 AG051426, and the National Institute on Minority Health and Health Disparities under Award Number U54 MD012388. The contents of this publication are solely the responsibility of the authors and do not necessarily represent the official views of these institutes and offices. The funders had no role in study design, data collection and analysis, decision to publish or preparation of the manuscript.

## Author contributions

Dakota D. Witzel: Study design, study conceptualization, data analysis, manuscript writing. Eric S. Cerino: Study design, study conceptualization, data analysis, manuscript writing. Robert S. Stawski: Study design, study conceptualization, data collection, manuscript writing, and reviewing. Gillian Porter: Data analysis, manuscript writing, and reviewing. Amanda D. Black: Data analysis, manuscript writing, and reviewing. Raechel A. Livingston: Data analysis, manuscript writing, and reviewing. Jonathan Rush: Data analysis, manuscript writing, and reviewing. Jacqueline Mogle: Study design, study conceptualization, data collection, manuscript writing, and reviewing. Susan T. Charles: Study design, study conceptualization, manuscript writing, and reviewing. Jennifer R. Piazza: Study design, study conceptualization, manuscript writing,g and reviewing. David M. Almeida: Study design, study conceptualization, data collection, manuscript writing, and reviewing.

## Competing interests
The authors declare no competing interests.
