## [Transparent Peer Review file · Communications Psychology]

Daily Association Between Perceived Control and Resolution of Daily Stressors Strengthens Across a Decade of Adulthood

Corresponding Author: Dr Eric Cerino

Version 0:

Decision Letter:

Dear Dr Cerino,

Thank you for your patience during the peer-review process. Your manuscript titled "Associations Between Perceived Control and Resolution of Daily Stressors Across a Decade of Adulthood" has now been seen by 2 reviewers, and I include their comments at the end of this message. They find your work of interest but raised some important points. We are interested in the possibility of publishing your study in Communications Psychology, but would like to consider your responses to these concerns and assess a revised manuscript before we make a final decision on publication.

We therefore invite you to revise and resubmit your manuscript, along with a point-by-point response to the reviewers. Please highlight all changes in the manuscript text file.

Editorially, we consider it important that you address the conceptual, methodological, and analytic concerns of the reviewers. We request that you conduct all necessary analyses to address the reviewers' concerns. In cases where the concerns cannot be fully resolved, please expand on your limitations section and explain how these concerns influence the interpretation of your findings. The reviewers provide recommendations for streamlining the introduction, as you do so, we ask that you clearly indicate where the work is exploratory and where you have a priori hypotheses, likewise please use care to avoid causal language where no causal evidence is available. Please ensure data and code are available for peer review upon resubmission.

I am attaching an Editorial Requests Table that details critical reporting requirements for the revised manuscript. Please attend to each item and ensure your manuscript is fully compliant. If your revised manuscript is not aligned with these requests on major issues, such as those concerning statistics, it may be returned to you for further revisions without re-review.

Please submit the following items:

- Revised manuscript
- Point-by-point response to the referees' comments
- Cover letter (as a separate document)
- <https://www.nature.com/documents/nr-reporting-summary.zip>>Nature Research Reporting Summary
- <https://www.nature.com/documents/nr-editorial-policy-checklist.pdf>>Editorial Policy Checklist

- Completed Editorial Request Table (attached).

via this link: Link Redacted .

Additional guidance is available in our style and formatting guide Communications Psychology formatting guide.

Best regards,

Nida Ali

Nida Ali, PhD
Editorial Board Member
Communications Psychology
orcid.org/0000-0003-4921-7940

REVIEWER EXPERTISE:

Reviewer #1, Stress resonance
Reviewer #2, Control, lifespan development

REVIEWER REPORTS:

Reviewer #1 (Remarks to the Author):

The present longitudinal study tested for associations between perceived control and resolution of daily stressors across adulthood. The authors found that people experiencing greater stressor control were more likely to resolve their stressors. Further, the link between stressor control and resolution became stronger at the 10-year follow-up. The research question is novel and a major strength of the study portrays the repeated assessment of perceived control and stressor resolution in a large sample of N=1,766 participants. The manuscript is well-written, the analyses appropriate. A few aspects deserve clarification:

Intro:

- A figure of the daily stress process model, which is central to the manuscript, would be helpful. This could highlight how the current study aims to expand components.
- An explanation on why resolution is a stressor appraisal is missing.
- While extensive arguments are provided for the main hypotheses, the aspect whether the strength of association changes with age could deserve more attention. No specific hypotheses are phrased, it seems to be exploratory. Could this be justified?
- I would assume that the severity of stressors plays a major role in affecting both perceived control and resolution. It would be nice if the authors could integrate this aspect, or if not deemed suitable, explain why. This also matters with regard to the discussion.

Methods:

- What was the rationale for dichotomizing daily stressor resolution?
- The analytic strategy is difficult to follow, particularly from ll. 190 on (e.g., "...within-person moderator of within-person associations and between-person associations in Model 3"). It could help to add the specific question that will be addressed?

Results:

- How many stressors were experienced on average per wave of data collection?
- How were covariates related to perceived control and resolution?

- Can you add effect sizes?

Discussion:

- The formulation of the first phrase may be reconsidered to avoid implying that perceived control is not a resource in younger age.

- The limitation of a potential bidirectionality of associations between control and resolution is shortly mentioned. Yet, this may be considered a major issue that would deserve a deeper reflection in the discussion.

- The authors highlight that one of the study's strength is the intensive longitudinal design. They further mention that disentangling the temporal effects would be critical. Wouldn't it be possible based on the present data to examine temporal effects? If not, can you justify why when reflecting on the limitations?

Reviewer #3 (Remarks to the Author):

Associations Between Perceived Control and Resolution of Daily Stressors Across a Decade of Adulthood

I want to note for transparency that I have previously reviewed this paper for another journal. After consulting with the editorial team, this journal still requested my review. Given that almost no substantive changes were made since my last review, I am providing my assessment here again.

This paper examines the cross-sectional and longitudinal association of stressor control and stressor resolution across adulthood using a dataset widely utilized in the stress-control literature. I really wanted to like this paper, as the topic is compelling and the NSDE dataset provides a solid foundation. However, it left me with more questions than answers. The combination of the design, the specific research question, and the simplistic operationalization of stressor resolution raises numerous questions about directionality and the robustness of the conclusions. Furthermore, the conceptual reasoning on age differences and longitudinal changes of the outcome feels underdeveloped. These challenges made it difficult to fully engage with the findings and their implications.

- I find the introduction to be somewhat lengthy and lacking in clear structure. Given that the focus is on stressor resolution, I would have expected a more direct and specific discussion on appraisal in the context of stressor resolution. A clearer and more focused introduction would better set the stage for the study's objectives.

- The concept of "Resolution of daily stressors" is central to this study, but it would be helpful to provide more depth regarding what is meant by "resolution." As this is the main outcome, I am particularly interested in understanding how stressor resolution is conceptualized in the established literature—does it encompass both practical and emotional aspects, or is it limited to one? The current item ("Was the stressor resolved? Yes/No") seems somewhat simplistic given the complexity of stress processes. Could the authors elaborate on the validity of this item? Specifically, I would like to know if any validation studies or psychometric analyses support the use of this measure to resolution. What does resolution actually mean? This additional information would help to better understand the robustness of the outcome.

- Relatedly, the definition provided for "resolution of daily stressors" includes both the cessation of the stressor and the emotional downregulation that follows. However, based on the actual item used ("Was the stressor resolved? Yes/No"), it is unclear how the emotional aspect of stressor resolution is captured here. It is possible that a stressor might be resolved practically (e.g., picking up the kids from school), but the emotional response may still persist or be unresolved. Could the authors clarify how emotional resolution is measured or accounted for in this context? If the emotional component is assumed rather than explicitly measured, it might be helpful to revise the definition to better align with the operationalization of the variable.

- The nature of stressors may vary significantly in their level of resolvability. For instance, a health-related stressor like arthritis pain or migraine may not be fully resolvable but is also age dependent. While younger individuals might experience acute stressors that can often be resolved, older adults might face chronic conditions that require ongoing management rather than a clear resolution. I missed a discussion of literature on resolvability and empirical work on daily stress beyond stressor control in the introduction of the paper.

- The manuscript states e.g., "Within persons, we hypothesize that individuals will be more likely to resolve their stressors on days when their stressor control is higher than usual." However, given that both stressor control and stressor resolution are measured at the same time at the end of each day, it is unclear how the authors infer the causal direction—that is, that stressor control leads to resolution rather than successful stressor resolution increasing perceptions of control over a stressor. The simultaneous measurement makes it difficult to disentangle the directionality between these variables. From conceptual perspective, a clearer justification for why stressor control is presumed to drive stressor resolution rather than the other way around (i.e., resolvability or actual resolution leading to an increased sense of control) would have been informative.

- Relatedly, the phrasing used in parts of the manuscript suggests a causal interpretation of the findings (e.g., "[...] perceived control may serve as a psychosocial resource for promoting [...]"; "This sensitivity [...] analysis found stressor control to be a resource for promoting resolution regardless of stressor type."). Given that the analysis is regression-based and also the

simultaneous end of the day measurement, such causal/directional language may overstate the nature of the findings. I recommend rephrasing these statements throughout the ms to better reflect associative relationships rather than causality, unless more causal inference techniques are employed. For example, using terms like "was associated with" or "linked to" would be more appropriate given the methods used. This adjustment will help avoid misinterpretation of the study's conclusions.

- Are all stressor domains equally likely to be resolved and does this vary by age? Independent of control. Please clarify?

- In later life, the urgency to resolve stressors may be less dictated by external circumstances, such as work /family life, which often impose immediate demands. Without these external factors, older adults might not "need" to resolve stressors right away, as their environment may allow more flexibility in how they respond. This aspect is missing from the text and is crucial for a comprehensive understanding of age-related differences in stressor resolution and its correlates.

- The ms would benefit from greater clarification on the concept of stressor resolution and how it may evolve into stressor chronicity. Specifically, if a stressor remains unresolved over time, does it eventually take on characteristics of a chronic stressor? It would be helpful for the authors to discuss whether the persistence of unresolved stressors over multiple days, as captured by the daily diary design, implies a transition from an acute, resolvable stressor to one that resembles a chronic stressor. In other words, does it matter if I carry an unresolved stressor through the week or if I have new (un)resolved stressors?

- As noted in the ms, the code for the analysis is available upon request. I would appreciate it if you could share the code, as I am particularly interested in understanding some modeling aspects.

- Some statements seem oversimplified, e.g. regarding perceived control across the lifespan. For example, the statement "Indeed, people's ability to independently exert control over their environment tends to decline with aging (Heckhausen et al., 2010; Lachman et al., 2009)." may overlook research that has demonstrated stability of perceived control into old age. Including references that illustrate both perspectives—decline versus stability—would provide a more nuanced and balanced view, acknowledging the diversity of findings in this area.

- The MIDUS/NSDE dataset allows for more detailed coding of education, e.g. year of education, yet it has been dichotomized here (0 = high school or less, 1 = some college or more). I recommend avoiding dichotomization, as it oversimplifies the construct and leads to a loss of valuable information.

- The authors state that "results from the current study clarify that individual differences in stressor control are useful in understanding who is more likely to resolve their stressors [...]." However, this conclusion seems rather intuitive, as it is generally expected that individuals with higher perceived control over a stressor are more likely to resolve stressors. I recommend expanding on this by addressing the "so what" aspect: Do we know that increasing control will actually help to solve a stressor or do we only know that stressors that are in fact more manageable, are also those stressors that are more likely to be resolved? Providing more context on why this matters would add value to the interpretation of the results.

If you experience problems in linking your ORCID, please contact the Platform Support Helpdesk.

Version 1:

Decision Letter:

Dear Dr Cerino,

Your manuscript titled "Associations Between Perceived Control and Resolution of Daily Stressors Across a Decade of Adulthood" has now been seen by our reviewers, whose comments appear below. In light of their advice I am delighted to say that we are happy, in principle, to publish a suitably revised version in Communications Psychology.

We therefore invite you to revise your paper one last time to address the remaining concerns of our reviewers and a list of editorial requests. At the same time we ask that you edit your manuscript to comply with our format requirements and to maximise the accessibility and therefore the impact of your work.

EDITORIAL REQUESTS:

SUBMISSION INFORMATION:

OPEN ACCESS:

*** TRANSPARENT PEER REVIEW:** Communications Psychology uses a transparent peer review system. On author request, confidential information and data can be removed from the published reviewer reports and rebuttal letters prior to publication. If you are concerned about the release of confidential data, please let us know specifically what information you would like to have removed. Please note that we cannot incorporate redactions for any other reasons.

*** CODE AVAILABILITY:** All Communications Psychology manuscripts must include a section titled "Code Availability" at the end of the methods section. We require that the custom analysis code supporting your conclusions is made available in a publicly accessible repository at this stage; please choose a repository that generates a digital object identifier (DOI) for the code; the link to the repository and the DOI must be included in the Code Availability statement. Publication as Supplementary Information will not suffice.

*** DATA AVAILABILITY:**

Link Redacted

Best regards,

Jennifer Bellingtier

Jennifer Bellingtier, PhD
Senior Editor
Communications Psychology

REVIEWER EXPERTISE:
Reviewer #1, Stress resonance

REVIEWERS' COMMENTS:

Reviewer #1 (Remarks to the Author):

The authors have adequately addressed all my comments.

Their answers are thorough and well-argued. In my opinion, the manuscript reflects those changes and will make a nice addition to the field.

The language could still be improved, and I would advise the authors to check again carefully the manuscript to facilitate better comprehensibility. The following typos need to be fixed, e.g.,: "domain-specific type of control and a may" (p. 2), "when stressors are resolved" (p. 6), research question 3: "moderated by longitudinal changes" -> instead of longitudinal changes, time or assessment wave would be more accurate (appears later on p. 11 again); "with children and (Lachman & Weaver, 1998)" (p 18); "An important future direction related to this is in the" (p. 20), "days may may accumulate" (p. 21).

Reviewer Comments to Author (with revisions indicated by *italics*)

Reviewer 1

The present longitudinal study tested for associations between perceived control and resolution of daily stressors across adulthood. The authors found that people experiencing greater stressor control were more likely to resolve their stressors. Further, the link between stressor control and resolution became stronger at the 10-year follow-up.

The research question is novel and a major strength of the study portrays the repeated assessment of perceived control and stressor resolution in a large sample of N=1,766 participants. The manuscript is well-written, the analyses appropriate. A few aspects deserve clarification:

Introduction:

1. A figure of the daily stress process model, which is central to the manuscript, would be helpful. This could highlight how the current study aims to expand components.
 - *Thank you for this suggestion. We now include a schematic diagram on associations between control and resolution of daily stressors embedded in the daily stress process model (Figure 1; see below). We included a dotted box around the control and resolution components of the model to communicate our emphasis on these two stressor appraisals in the current study. We also included a citation for Almeida (2024) to refer readers to additional information on the daily stress process model for a more comprehensive depiction of each component. We intentionally greyed out the components of the model not under investigation in the current study to draw attention to our focus on stressor appraisals (i.e., control, resolution).*

2. An explanation on why resolution is a stressor appraisal is missing.
 - *We have provided additional detail on the definition of resolution and its operationalization as a stressor appraisal in the Introduction and Method sections. In the Introduction, we clarify that, “Resolution of daily stressors is defined as the subjective appraisal that a stressor is no longer ongoing (Witzel & Stawski, 2021).” In the Method section, we added the following text to clarify that resolution is part of the participant’s general appraisal of the stressful situation. It now*

reads, “resolution referred to a subjective appraisal that each participant interprets for themselves when characterizing the stressor they experienced (e.g., the participant appraises their stressor as resolved and answers the interview question with “Yes”).”

3. While extensive arguments are provided for the main hypotheses, the aspect of whether the strength of association changes with age could deserve more attention. No specific hypotheses are phrased, it seems to be exploratory. Could this be justified?
 - *We agree that the original submission did not expand upon reasons why we may expect to see age differences or longitudinal moderation of associations between control and resolution. In the revised submission, we now clarify specific research questions and hypotheses for baseline age differences (Research Question 2) and longitudinal moderation (Research Question 3). We provide justification for these questions and hypotheses in the Introduction with new writing on the potential advantages for allocating control over stressors in later adulthood and as individuals grow older. There is now a new paragraph in the stressor control section reviewing age patterns in control and the potential for links between control and resolution of daily stressors to change in magnitude as people grow older and shift in their priorities, activities, and resources (Heckhausen et al., 2010). Further, the stressor resolution paragraphs now report age differences examined in past research (e.g., older age related to more interpersonal stressor resolution).*
 - *The second and third research questions now read, “Are associations between control and resolution of daily stressors moderated by baseline age? We examine control by baseline age interactions to evaluate whether the positive association between control and resolution is stronger among comparatively older adults. Are associations between control and resolution of daily stressors moderated by longitudinal changes? We examine control by wave of assessment interactions to evaluate whether control-resolution associations change across 10 years of follow-up.”*
4. I would assume that the severity of stressors plays a major role in affecting both perceived control and resolution. It would be nice if the authors could integrate this aspect, or if not deemed suitable, explain why. This also matters with regard to the discussion.
 - *Thank you for this suggestion. We agree that stressor severity could play a significant role in control and resolution of daily stressors and think this is an important component of the stress process to account for in analyses. Per this recommendation, we reran each model covarying for stressor severity. Primary findings held when additionally adjusting for the severity of daily stressors. We now include this sensitivity analysis in Supplemental Material (Supplementary Table 2).*

Methods:

5. What was the rationale for dichotomizing daily stressor resolution?
 - *We dichotomized stressor resolution primarily due to the dichotomous nature of the protocol question “Is the situation resolved? Yes or no?” aligned with our dichotomization of 0=no stressors resolved, 1=at least 1 resolved stressor. However, we recognize that this work is limited to the single question in the current study’s protocol and thus does not comprehensively*

cover all features of the resolution appraisal. We have added to the Future Directions section a call for research to examine additional features of the resolution appraisal. The Future Directions section now reads, “The present study focused on an individual’s subjective appraisal of whether their stressors were resolved or not. Because resolution was limited to a single yes/no dichotomous question of whether the situation was resolved on that particular day, future work should examine additional features of the resolution process, such as when the stressor was resolved, how it was resolved, and the possible emotional responses associated with the resolution experience.”

6. The analytic strategy is difficult to follow, particularly from ll. 190 on (e.g., “...within-person moderator of within-person associations and between-person associations in Model 3”). It could help to add the specific question that will be addressed?
 - *Thank you for the opportunity to improve the clarity and readability of our analytic strategy. We have rewritten our Research Questions/Hypotheses in the Introduction and the Analytic Strategy to better align the analytical components to the primary aims of the current study. The Analytic Strategy now repeats each research question before identifying which parameter in the provided equation maps onto the effect of interest.*
 - *The Analytic Strategy now reads, “To answer research question 1 (Are control and resolution of daily stressors associated both within-persons over time and between-persons?), we regressed daily stressor resolution on within-person stressor control (β_{3ij}) and between-person stressor control (γ_{011}). To answer research question 2 (Are associations between control and resolution of daily stressors moderated by baseline age?), we added two interaction terms. Specifically, we added the within-person stressor control*baseline age two-way interaction (γ_{301}) and the between-person stressor control*baseline age two-way interaction (γ_{006}) as predictors of stressor resolution in Model 2. To answer research question 3 (Are associations between control and resolution of daily stressors moderated by longitudinal changes?), we included two additional interaction terms. Specifically, we added the within-person stressor control*wave of assessment two-way interaction (δ_{31i}) and the between-person stressor control*wave of assessment two-way interaction (γ_{011}) as predictors of stressor resolution in Model 3.”*

Results:

7. How many stressors were experienced on average per wave of data collection?
 - *Stressors were experienced on 40% of all available days in Wave 2 and 39% of all available days in Wave 3. We include this information in the Measures section under “Daily stressor control” with the following sentence, “For each stressor reported (occurring on 40% of all available days in Wave 2, 39% of all available days in Wave 3), participants answered the question, “How much control did you have over the situation?” on a 4-point Likert-type scale (0=none at all, 1=a little, 2=some, 3=a lot).*
8. How were covariates related to perceived control and resolution?
 - *Bivariate correlations for covariates, perceived control, and resolution are provided in Table 1 for Wave 2 (below the diagonal) and Wave 3 (above the diagonal). We have now added descriptions of significant correlations with covariates to the Results section.*

- *After the paragraph on descriptive statistics and variance decompositions for control and resolution are provided, a second paragraph now reads, “At Wave 2, significant bivariate correlations with covariates showed higher stressor control was reported more by men (compared to women; $r=-0.11$, $p<.001$), non-White respondents (compared to White respondents; $r=0.09$, $p<.001$), and people with fewer numbers of stressors on stress days ($r=-0.06$, $p<.01$). Further, stressor resolution at Wave 2 was significantly related to education and number of stressors such that people with less than some college ($r=-0.09$, $p<.001$) and fewer numbers of stressors on stress days ($r=-0.07$, $p<.01$) reported higher percentage of days when at least one stressor was resolved. At Wave 3, significant bivariate correlations showed stressor control was reported more by men (compared to women; $r=-0.10$, $p<.001$). Further, stressor resolution at Wave 3 was significantly related to education and number of stressors such that people with less than some college ($r=-0.11$, $p<.001$) and fewer numbers of stressors on stress days ($r=-0.06$, $p<.05$) reported higher percentage of days when at least one stressor was resolved.”*

9. Can you add effect sizes?

- *In the original submission, we leveraged the odds ratio metric as a useful measure of the magnitude of associations between control and resolution. This is consistent with past research identifying odds ratios as a metric of effect size (Maher, Markey, & Ebert-May, 2013). In the first paragraph of the Discussion section, we discuss the longitudinal moderation as the percentage increase in the odds of resolving stressors on days when control was higher than average at baseline (56% increase in odds) vs. ten years later (89% increase in odds) and interpreted the odds ratio. In the revised submission, we added predicted probabilities ($\frac{\text{Odds}}{1+\text{Odds}}$) for significant main effects and interaction effects to further aid in the interpretability of significant main effects and interaction effects.*
- *The Analytic Plan now reads, “To further aid in the interpretability of effects, we also calculated the predicted probability ($\frac{\text{Odds}}{1+\text{Odds}}$) of resolving at least one stressor given a 1-unit increase in stressor control compared to average control.” Further, predicted probabilities for significant main effects and interaction effects are included in the Results section under subsections “**Main effects.**” And “**Longitudinal moderation.**”*

Discussion:

10. The formulation of the first phrase may be reconsidered to avoid implying that perceived control is not a resource in younger age.
- *Thank you for catching this misleading statement. We have replaced the words “as people grow older” with “across the adult lifespan” in this first sentence and revised the wording to align with Reviewer 2’s comments as well. The first sentence of the Discussion now reads, “Results from this study indicated that perceived control is a psychosocial correlate of stressor resolution and an important appraisal resource for daily stress process across the adult lifespan.”*
11. The limitation of a potential bidirectionality of associations between control and resolution is shortly mentioned. Yet, this may be considered a major issue that would deserve a deeper reflection in the discussion.

- *We agree that this is an important point that warrants more explanation in the Limitations and Future Directions section. We have revised the manuscript to identify this as an important limitation and expand on reasons why we were unable to evaluate control and resolution over the same stressors across multiple days (specific text provided below in response to Reviewer 1 Point 12).*
12. The authors highlight that one of the study's strengths is the intensive longitudinal design. They further mention that disentangling the temporal effects would be critical. Wouldn't it be possible based on the present data to examine temporal effects? If not, can you justify why when reflecting on the limitations?
- *Thank you for raising this point and for the opportunity to clarify study design considerations that prevented our capacity to evaluate the temporal effects from one day to the next. Respondents reset their reporting of daily stressors each day so performing lagged effects analyses would not confidently map on control and resolution over the same stressors from one day to the next. We have added additional reflection and justification in the Limitations and Future Directions. This section now reads, "Further, it was not possible to evaluate control and resolution over the same stressors across multiple days (respondents reset their reporting of daily stressors each day). Thus, an important limitation of the current study is its inability to evaluate temporal effects of control on resolution of specific stressors from one day to the next. Future research should address this limitation with more momentary assessments within days (e.g., ecological momentary assessment designs) to examine whether increases in control lead to subsequent resolution of stressors and/or if people increase their perception of control in response to resolving a stressor. Disentangling the temporal effects of resources, control, and resolution will become crucial for understanding how and when to intervene in daily stress processes."*

Reviewer 3

I want to note for transparency that I have previously reviewed this paper for another journal. After consulting with the editorial team, this journal still requested my review. Given that almost no substantive changes were made since my last review, I am providing my assessment here again.

This paper examines the cross-sectional and longitudinal association of stressor control and stressor resolution across adulthood using a dataset widely utilized in the stress-control literature. I really wanted to like this paper, as the topic is compelling and the NSDE dataset provides a solid foundation. However, it left me with more questions than answers. The combination of the design, the specific research question, and the simplistic operationalization of stressor resolution raises numerous questions about directionality and the robustness of the conclusions. Furthermore, the conceptual reasoning on age differences and longitudinal changes of the outcome feels underdeveloped. These challenges made it difficult to fully engage with the findings and their implications.

- *Thank you for your willingness to provide constructive feedback on an earlier version of this paper and the current submitted paper. We are grateful for the points you've raised and feel the resulting revisions to the paper have strengthened its' conceptual and methodological contributions to the literature.*

1. I find the introduction to be somewhat lengthy and lacking in clear structure. Given that the focus is on stressor resolution, I would have expected a more direct and specific discussion on appraisal in the context of stressor resolution. A clearer and more focused introduction would better set the stage for the study's objectives.
 - *In the revised submission, we worked toward finding a balance between clear structure and comprehensive coverage of each primary component of the present study. In addition to our focus on stressor resolution, we similarly focus on the role of stressor control as a unique correlate of stressor resolution. We see the value of the present study as empirical support of control and resolution as integrated components of the daily stress process model (e.g., within- and between-person associations) with relevance for adult development and aging (longitudinal moderation of daily associations across ten years of follow-up). With this in mind, we reworked the Introduction to ensure literature was represented for each of these components.*

2. The concept of "Resolution of daily stressors" is central to this study, but it would be helpful to provide more depth regarding what is meant by "resolution." As this is the main outcome, I am particularly interested in understanding how stressor resolution is conceptualized in the established literature—does it encompass both practical and emotional aspects, or is it limited to one? The current item ("Was the stressor resolved? Yes/No") seems somewhat simplistic given the complexity of stress processes. Could the authors elaborate on the validity of this item? Specifically, I would like to know if any validation studies or psychometric analyses support the use of this measure to resolution. What does resolution actually mean? This additional information would help to better understand the robustness of the outcome.
 - *Consistent with our response to Point 3 below, we revised sections of the manuscript to clarify the operationalization of stressor resolution in past literature and the current study. We reviewed past literature that broadly defined resolution as a stressor that is no longer ongoing. The Introduction section on stressor resolution now reads, "The operationalization of stressor resolution varies widely across studies, from reporting the exact date (year/month) of the exposure and concrete end (Harnish et al., 2000) to a multi-indicator item determining conflict resolution (Henning, 2004) to a dichotomous Yes/No response for resolution of specific daily stressors at the end of each day of daily diary (Witzel & Stawski, 2021). Although research has utilized multiple indicators for resolution status over time, comparatively less attention has been dedicated to resolution of stressors that operate on a daily basis."*
 - *While we are not aware of any formal psychometric evaluation of the stressor resolution variable used in the present study, we do highlight past research demonstrating its validity as a correlate of decreased affective reactivity and residue associated with interpersonal stressors (marking its relevance for downregulation of emotions and importance for daily well-being). Specifically, we report literature showing that affective reactivity and residue associated with interpersonal stressors was attenuated or even extinguished when stressors are resolved (Witzel & Stawski, 2021).*
 - *Notably, and in line with other reviewer comments, one limitation of the daily inventory of stressful experiences is the single-item nature of resolution. As such, this is an initial investigation into these novel indicators of daily stress processes. We have now added language*

to reflect this in the limitations and future directions sections within the Discussion (see response to point 3).

3. Relatedly, the definition provided for "resolution of daily stressors" includes both the cessation of the stressor and the emotional downregulation that follows. However, based on the actual item used ("Was the stressor resolved? Yes/No"), it is unclear how the emotional aspect of stressor resolution is captured here. It is possible that a stressor might be resolved practically (e.g., picking up the kids from school), but the emotional response may still persist or be unresolved. Could the authors clarify how emotional resolution is measured or accounted for in this context? If the emotional component is assumed rather than explicitly measured, it might be helpful to revise the definition to better align with the operationalization of the variable.
 - *We appreciate the reviewer identifying the need to clarify what is being assessed. The current study operationalizes stressor resolution as the subjective appraisal that a specific stressor event is no longer ongoing. There is no emotional response included in how we measured stressor resolution. The confusion may have been caused by our original manuscript's text saying "Resolution of daily stressors is defined as the subjective indication that a stressor is no longer ongoing and is critical for the downregulation of emotions". We attempted to couple the relevance of resolution for emotional downregulation in the same sentence as the definition. In the revised submission, we have broken this up into two sentences so that resolution can be more clearly defined as the subjective appraisal that a stressor is no longer ongoing. Then, in the subsequent sentence, we can introduce its relevance for health and well-being due to its role in the downregulation of emotions. Key distinctions are now made in the Introduction, Method, and Discussion.*
 - *The Introduction now reads, "Resolution of daily stressors is defined as the subjective appraisal that a stressor is no longer ongoing (Witzel & Stawski, 2021). This stressor appraisal may be critical for the downregulation of emotions (Harnish et al., 2000), with recent work demonstrating decreases in affective responses when daily stressors are resolved (Witzel & Stawski, 2021)."*
 - *The Method now reads, "Resolution referred to a subjective appraisal that each participant interprets for themselves when characterizing the stressor they experienced (e.g., the participant appraises their stressor as resolved and answers the interview question with "Yes")."*
 - *The Discussion now reads, "The present study focused on an individual's subjective appraisal of whether their stressors were resolved or not. Although we ask about whether the stressor itself has been resolved, this question still raises the possibility that people are thinking about what happened and may have lingering emotions in response. The present study was a first step in assessing resolution. Future research can examine additional features of the resolution process, such as when the stressor was resolved, how it was resolved, and the possible emotional responses associated with the resolution experience."*
4. The nature of stressors may vary significantly in their level of resolvability. For instance, a health-related stressor like arthritis pain or migraine may not be fully resolvable but is also age dependent. While younger individuals might experience acute stressors that can often be resolved, older adults might face chronic conditions that require ongoing management rather

than a clear resolution. I missed a discussion of literature on resolvability and empirical work on daily stress beyond stressor control in the introduction of the paper.

- *We agree that stressors vary in their level of resolvability. We now review past research that has examined resolvability of different types of focal stressors. The Introduction section on stressor resolution now reads, “Harnish and colleagues (2000) showed that the resolvability of focal stressors reported in the past year varies as a function of major stressor type. While median duration for focal stressors to be resolved was ~7 months across all types of stressors, transition stressors were resolved in the shortest amount of time, followed by interpersonal, illness, and role strain stressors. The present study extends this past work to evaluate resolution across daily domains of interpersonal, work, home, and network stressors.”*
 - *Regarding the missing empirical work on daily stress beyond stressor control, we have incorporated additional research on age differences in stressor exposure, severity, and reactivity in the introduction “Indeed, daily stress research shows people in midlife often report more daily stressors and perceive their stressors as more severe than older adults (Almeida & Horn, 2004). Further, coordinated analysis of intensive repeated measurement studies shows age-related reductions in stressor reactivity (Stawski et al., 2019).”*
 - *Further, the reviewer raises an important point on the nature of stressors potentially varying in their level of resolvability. We are confident in the conclusions drawn from the present study’s findings, given our sensitivity analyses demonstrating results were robust across different types of stressors and when adjusting for perceived severity of daily stressors.*
5. The manuscript states e.g., "Within persons, we hypothesize that individuals will be more likely to resolve their stressors on days when their stressor control is higher than usual." However, given that both stressor control and stressor resolution are measured at the same time at the end of each day, it is unclear how the authors infer the causal direction—that is, that stressor control leads to resolution rather than successful stressor resolution increasing perceptions of control over a stressor. The simultaneous measurement makes it difficult to disentangle the directionality between these variables. From conceptual perspective, a clearer justification for why stressor control is presumed to drive stressor resolution rather than the other way around (i.e., resolvability or actual resolution leading to an increased sense of control) would have been informative.
- *Thank you for raising this important point. We have made three key revisions to the manuscript to adjust our language to align with the correlational design and justify our use of resolution as the target outcome and control as the target predictor in our regression-based models. First, we revised language throughout to emphasize language communicating associations (e.g., “associations between”, “correlate”) instead of talking about control as a resource for resolution (and further bring this point up in the Limitations and Future Directions section. Second, we formally included a schematic diagram of the daily stress process model where we include a bidirectional arrow between control and resolution communicating the correlational findings we test. Third, we added text to the Present Study section that justifies why we conceptualize control as a modifiable psychosocial actor with relevance as a predictor of a meaningful endpoint in a person’s day (e.g., whether the stress someone reported was settled or*

not). The Present Study section now reads, “Although our study design is correlational, we conceptualize stressor resolution as the target outcome and stressor control as the focal predictor to identify a potentially modifiable psychosocial correlate (i.e., control) of a meaningful outcome at the end of a day (whether a stressor has been resolved or not). Indeed, given resolution’s relevance for emotional downregulation in past research (e.g., Harnish et al., 2000; Witzel & Stawski, 2021), it is important to identify daily factors that may be correlated with increased likelihood of resolving one’s stress. Further, past research has examined resolution as an outcome regressed on psychosocial predictors (e.g., Brennan et al., 2006)”.

6. Relatedly, the phrasing used in parts of the manuscript suggests a causal interpretation of the findings (e.g., “[...] perceived control may serve as a psychosocial resource for promoting [...]”); “This sensitivity [...] analysis found stressor control to be a resource for promoting resolution regardless of stressor type.”). Given that the analysis is regression-based and also the simultaneous end of the day measurement, such causal/directional language may overstate the nature of the findings. I recommend rephrasing these statements throughout the ms to better reflect associative relationships rather than causality, unless more causal inference techniques are employed. For example, using terms like “was associated with” or “linked to” would be more appropriate given the methods used. This adjustment will help avoid misinterpretation of the study’s conclusions.
 - *Thank you for ensuring the takeaways from the present study are not misinterpreted and the manuscript’s writing is aligned with the correlational nature of the analyses. We have revised the manuscript’s language throughout each section by removing instances where we write perceived control may serve as a resource for stressor resolution. Instead, we now write that control is associated with resolution and use the term “correlate” in subsection text and subheaders (e.g., Discussion subheader “**Control as a Psychosocial Correlate of Daily Stressor Resolution**”). This rewording aligns with existing writing in the manuscript’s limitations section emphasizing that “The within-person associations in the current study’s daily diary design are correlational.”*
7. Are all stressor domains equally likely to be resolved and does this vary by age? Independent of control. Please clarify?
 - *This is an important question that we partially addressed in the original submission’s sensitivity analysis examining associations between control and resolution among different types of stressors (Supplementary Table 4 in revised manuscript; labeled Supplementary Table 2 in the original submission). New to this revision, we have also added Supplementary Table 5 to provided additional descriptive statistics on resolution across all stressor domains so that the manuscript provides the percentage of days when at least one stressor was resolved across stressor days for each type of stressor. Arguments, avoided arguments, and work stressors (between 67% and 69% of these stressors were reported as resolved at Waves 2 and 3) were resolved more often than home stressors (57% of home stressors reported as resolved at Waves 2 and 3) and network stressors (41% and 44% of network stressors reported as resolved at Wave 2 and Wave 3, respectively). Supplementary Table 4’s multilevel models facilitate examination of*

potential age differences in resolution independent of control (or over and above the influence of control) across stressor domains using the “Age at Baseline” parameter. Older age at baseline was associated with greater stressor resolution for arguments (OR=1.01, 95% CI: 1.003–1.03, $p < .05$) and avoided arguments (OR=1.01, 95% CI: 1.002–1.02, $p < .05$), but not for work stressors (OR=0.99, 95% CI: 0.98–1.01, $p > .05$), home stressors (OR=1.001, 95% CI: 0.99–1.01, $p > .05$), and network stressors (OR=0.997, 95% CI: 0.99–1.01, $p > .05$).

8. In later life, the urgency to resolve stressors may be less dictated by external circumstances, such as work/family life, which often impose immediate demands. Without these external factors, older adults might not “need” to resolve stressors right away, as their environment may allow more flexibility in how they respond. This aspect is missing from the text and is crucial for a comprehensive understanding of age-related differences in stressor resolution and its correlates.
 - *This is an interesting point that complements existing interpretations in the manuscript. We have added a new paragraph to the “**Longitudinal changes in the daily control-resolution association.**” subsection of the Discussion to highlight the potential for the longitudinal moderation to be due in part to shifts in social roles and external circumstances that may impact the need for stressor resolution as people grow older. The section now reads, Work and family demands earlier in adulthood may be characterized by more external circumstances (e.g., preparing for an external evaluation at work or helping your parent manage medications before their doctor’s appointment). As people grow older and shift into social roles comparatively less-dictated by external circumstances, their environments may allow for more flexibility to resolve the stressors using their own control resources. This potential heightened alignment between fewer external circumstances and greater capacity to allocate control over the stressors that matter to them may also be partially explaining why the within-person control and resolution link grew stronger ten years later.*
9. The ms would benefit from greater clarification on the concept of stressor resolution and how it may evolve into stressor chronicity. Specifically, if a stressor remains unresolved over time, does it eventually take on characteristics of a chronic stressor? It would be helpful for the authors to discuss whether the persistence of unresolved stressors over multiple days, as captured by the daily diary design, implies a transition from an acute, resolvable stressor to one that resembles a chronic stressor. In other words, does it matter if I carry an unresolved stressor through the week or if I have new (un)resolved stressors?
 - *These are interesting ideas that introduce an exciting line of future directions from this work. Related to our response to Reviewer 1 Points 11 and 12, there are study design limitations that unfortunately prevent us from examining specific stressors that remain unresolved over consecutive days (respondents reset their reporting of daily stressors each day). However, we agree that the manuscript would benefit from additional reflection and discussion on what future research is needed to evaluate persistent unresolved stressors. We have added to our paper past work by Koffler and colleagues (2016) as one example of past research focused on exposure to different domains of stressors that could be relevant for informing research on the resolution of daily stressors. We have added to the Limitations section to explain that “it was not possible to evaluate control and resolution over the same stressors across multiple days (respondents reset*

their reporting of specific daily stressors each day). Thus, an important limitation of the current study is in its inability to evaluate temporal effects of control on resolution of specific stressors from one day to the next.” In the Future Direction section, we have also added, “One way to study chronicity of stressor exposures is to assess how often the same domains of stressors are reported (e.g., Koffer et al., 2016). Past research shows that the combination of low diversity of stressor exposure and high levels of stressor exposure in general are associated with elevated stressor reactivity and may reflect the chronicity of daily stressors (Koffer et al., 2016). Future measurement burst research that incorporates more comprehensive features of the resolution process for these stressor exposures embedded within macro-longitudinal follow-up can evaluate how persistent unresolved stressors over multiple days may accumulate over consecutive days and months to become a chronic strain.”

10. As noted in the ms, the code for the analysis is available upon request. I would appreciate it if you could share the code, as I am particularly interested in understanding some modeling aspects.
 - *We have included the code for all analyses presented in tables and figures as supplemental material in this resubmission. This is labeled Appendix B: Analysis Scripts).*

11. Some statements seem oversimplified, e.g. regarding perceived control across the lifespan. For example, the statement " Indeed, people’s ability to independently exert control over their environment tends to decline with aging (Heckhausen et al., 2010; Lachman et al., 2009)." may overlook research that has demonstrated stability of perceived control into old age. Including references that illustrate both perspectives—decline versus stability—would provide a more nuanced and balanced view, acknowledging the diversity of findings in this area.
 - *Thank you for this important point and opportunity to review additional work that more accurately depicts stability and change in control beliefs across the adult lifespan. We have revised this writing in the Discussion to include empirical examples of the nuanced literature on control in general and over specific life domains. This section now reads, “People’s ability to independently exert control over their environment generally tends to increase throughout younger adulthood, remain relatively stable in midlife, and then decrease in later adulthood (Cerino et al., 2023; Heckhausen et al., 2010; Lachman et al., 2009). Control over specific life domains, however, are more nuanced (e.g., for review, see Drewelies et al., 2019), with older age related to greater control over their work and finances, but less control over domains such as their sex life and relationships with children and (Lachman & Weaver, 1998). Further, certain types of control in daily life such as interpersonal stressor control stay stable across up to a 10-year follow-up (Cerino et al., 2024).*

12. The MIDUS/NSDE dataset allows for more detailed coding of education, e.g. year of education, yet it has been dichotomized here (0 = high school or less, 1 = some college or more). I recommend avoiding dichotomization, as it oversimplifies the construct and leads to a loss of valuable information.
 - *Thank you for this suggestion. In new sensitivity analysis, we reran each model covarying for the education level (range=1-12) variable instead of the dichotomous covariate used in primary*

analysis. Primary findings held when using education level instead of the dichotomous covariate. Thus, we retained our primary analyses and model components and included this sensitivity analysis in Supplemental Material (Supplementary Table 3).

- *The main text of the Results section now reads, “**Adjusting for education level.** We determined whether findings operated similarly when covarying for education level (range=1-12) instead of the dichotomous education covariate used in primary analysis (see Supplementary Table 3). Results revealed that primary findings held when covarying for education level instead of the dichotomous covariate.”*
13. The authors state that "results from the current study clarify that individual differences in stressor control are useful in understanding who is more likely to resolve their stressors [...]." However, this conclusion seems rather intuitive, as it is generally expected that individuals with higher perceived control over a stressor are more likely to resolve stressors. I recommend expanding on this by addressing the “so what” aspect: Do we know that increasing control will actually help to solve a stressor or do we only know that stressors that are in fact more manageable, are also those stressors that are more likely to be resolved? Providing more context on why these matters would add value to the interpretation of the results.
- *This is an important point for ensuring the findings we share in the current study hold relevance for our field moving forward. Although the main effects reported in the current study are rather intuitive and generally expected, we feel it is important to demonstrate these associations with empirical evidence, as there can be intuitive associations that are not held in a systematic, rigorous empirical testing. Thus, we prioritize the present scientific evaluation as our basis for drawing conclusions. In this way, the present study serves as a necessary foundation for lines of programmatic research to follow that intentionally incorporate control and resolution of daily stressors into larger empirical tests of the daily stress process model’s role in understanding health and well-being in daily life. We have added a paragraph to the Discussion section on “Control and Resolution as Integrated Components of the Daily Stress Process Model” to further demonstrate this value added. The section reads, “Although these were expected associations consistent with our hypotheses, it is important to formally demonstrate associations among a modifiable psychosocial resource like stressor control and the likelihood of resolving daily stressors to inform more personalized approaches to healthy aging (Hamburg & Collins, 2010). Indeed, the strengthening of control-resolution associations over time and consistency in findings across different types of stressors identifies the importance of studying control-resolution linkages as people grow older. Future efforts aimed at enhancing daily stress responses by increasing individuals’ capacity to resolve stressors may be effective not only for individuals with high levels of stressor control, but also for all adults when their momentary sense of control is elevated.*

August 11, 2025

Nida Ali, PhD
Editorial Board Member
Communications Psychology
Springer Nature
University of Vienna
Liebiggasse 5, Department of Clinical and Health Psychology
1010 Wien, Austria
Email: nida.ali@univie.ac.at

Dear Dr. Ali, Dr. Bellingtier, and Reviewers,

Thank you both for the opportunity to make a final revision to our manuscript entitled, '*Daily Association Between Perceived Control and Resolution of Daily Stressors Strengthens Across a Decade of Adulthood*'. My co-authors and I appreciate all comments made by you and Reviewer 1 in this recent round of review. We are writing this cover letter to confirm. All revisions to the manuscript are made in **red** text. Below are the specifications of how we addressed each comment made by you and the two reviewers.

Thank you for your time and consideration in evaluating the suitability of this revised manuscript for publication in *Communications Psychology*'s collection of papers on Intensive Longitudinal Designs in Psychology. If you have any questions, please contact me at Eric.Cerino@nau.edu. I look forward to hearing from you.

Sincerely,

--

Eric Cerino, PhD (He/Him/His)
Associate Professor of Psychological Sciences
College of Social and Behavioral Sciences
Northern Arizona University

cc: Dakota D. Witzel, PhD
Robert S. Stawski, PhD
Gillian Porter, PT
Amanda D. Black, BS
Raechel A. Livingston, BS
Jonathan Rush, PhD
Jacqueline Mogle, PhD
Susan T. Charles, PhD
Jennifer R. Piazza, PhD
David M. Almeida, PhD

Editorial Checklist Completed

- *Thank you for the comprehensive recommendations and assistance with ensuring our manuscript aligns with the formatting and style of Communications Psychology. We have addressed all editorial checklist requests.*

Reviewer Comments to Author (with revisions indicated by *italics*)

Reviewer 1

The authors have adequately addressed all my comments. Their answers are thorough and well-argued. In my opinion, the manuscript reflects those changes and will make a nice addition to the field.

The language could still be improved, and I would advise the authors to check again carefully the manuscript to facilitate better comprehensibility. The following typos need to be fixed, e.g.: "domain-specific type of control and a may" (p. 2), "when stressors are resolved" (p. 6), research question 3: "moderated by longitudinal changes" -> instead of longitudinal changes, time or assessment wave would be more accurate (appears later on p. 11 again); "with children and (Lachman & Weaver, 1998)" (p 18); "An important future direction related to this is in the" (p. 20), "days may may accumulate" (p. 21).

Introduction:

- *Thank you very much for all of your help and constructive recommendations to improve our paper. We have made all of these changes in the revised submission.*